# Morphological and Immunocytochemical Characterization of Paclitaxel-Induced Microcells in Sk-Mel-28 Melanoma Cells

**DOI:** 10.3390/biomedicines12071576

**Published:** 2024-07-16

**Authors:** Zane Simsone, Tālivaldis Feivalds, Līga Harju, Indra Miķelsone, Ilze Blāķe, Juris Bērziņš, Indulis Buiķis

**Affiliations:** 1Institute of Cardiology and Regenerative Medicine, The University of Latvia, Jelgavas Street 3, LV-1004 Riga, Latvia; talivaldis.freivalds@gmail.com (T.F.); roconis@gmail.com (J.B.); freivald@latnet.lv (I.B.); 2Department of Human Physiology and Biochemistry, Rīga Stradiņš University, Dzirciema Street 16, LV-1007 Riga, Latvia; indra.mikelsone@rsu.lv; 3Faculty of Medicine and Life Science, The University of Latvia, Jelgavas Street 1, LV-1004 Riga, Latvia; ilze.blake@lu.lv

**Keywords:** cancer biology, early detection, new paradigms, microcell, macrocell, sporosis, paclitaxel, SOX2, PCNA, caspase, Nanog

## Abstract

Biomarkers, including proteins, nucleic acids, antibodies, and peptides, are essential for identifying diseases such as cancer and differentiating between healthy and abnormal cells in patients. To date, studies have shown that cancer stem cells have DNA repair mechanisms that deter the effects of medicinal treatment. Experiments with cell cultures and chemotherapy treatments of these cultures have revealed the presence of small cells, with a small amount of cytoplasm that can be intensively stained with azure eosin, called microcells. Microcells develop during sporosis from a damaged tumor macrocell. After anticancer therapy in tumor cells, a defective macrocell may produce one or more microcells. This study aims to characterize microcell morphology in melanoma cell lines. In this investigation, we characterized the population of cancer cell microcells after applying paclitaxel treatment to a Sk-Mel-28 melanoma cell line using immunocytochemical cell marker detection and fluorescent microscopy. Paclitaxel-treated cancer cells show stronger expression of stem-associated ALDH2, SOX2, and Nanog markers than untreated cells. The proliferation of nuclear antigens in cells and the synthesis of RNA in microcells indicate cell self-defense, promoting resistance to applied therapy. These findings improve our understanding of microcell behavior in melanoma, potentially informing future strategies to counteract drug resistance in cancer treatment.

## 1. Introduction

Cancer cells are highly heterogeneous and polymorphic within tumor tissue [1,2]. The heterogeneity of cancer cells is described at both genetic and phenotypical levels [3,4]. Cancer resistance and stem cell-like properties, such as proliferation and self-renewal, have been linked to heterogenetic cells [3,5,6]. Like cancer cells, stem cells divide, proliferate, and self-renew uncontrollably. The cancer cell population with renewal abilities is a minor subpopulation. This subpopulation gains the ability to be resistant or tolerant to anticancer treatment [7,8]. Therapeutic resistance to cancer resistance is often associated with chemoresistance, which interferes with proliferation [9,10]. The chemotherapy drug doxorubicin (DOX) is a topoisomerase II inhibitor involved in DNA intercalation between base pairs, causing DNA breakage and inhibition of DNA and RNA synthesis. Doxorubicin is a type of chemotherapeutic drug used against different types of cancer. The expression of one of the cancer stem cell markers ALDH (aldehyde dehydrogenase) indicates the ability of cells to resist treatment by cytotoxic drugs, such as hydro-peroxy-cyclophosphamide (4-HC) and doxorubicin [11,12]. The other biomarker, CD44, has been discovered to positively modulate the nuclear factor of erythroid-2-like 2 (the main regulator of antioxidant genes) in DOX-resistant breast cancer cells [13,14]. At the same time, paclitaxel (PTX) is an antimitotic drug [15,16]. PTX is an antimicrotubular agent of Taxus brevifolia, derived from the taxane tree, that stops the G0/G1 and G2/M phases and inhibits replication and cytoskeletal rearrangement [17,18,19]. PTX also induces the generation of reactive oxygen species by improving the glucose mechanism [20,21]. In stemness research, in PC-3 cells resistant to paclitaxel, the expression levels of C-MYC and NANOG were significantly higher than in parental PC-3 cells [22]. As cell development is impaired, the cell should die. However, this is not always the case: cells survive and the tumor regrows and metastasizes.

In cancer, a cell subpopulation with high genetic heterogeneity creates an opportunity for the development of a resistant cell population. Cancer cell resistance could be inherited when tumor cells at the beginning are resistant to applied therapy [23,24]. Most of the time, this is explained by genetic resistance through natural selection. However, genetic mutations occur at random in single cells, suggesting the development of cancer [9,25]. Drug resistance is not associated with genetic alterations, which refer to functional adaptations [3,26]. Additionally, resistance could be acquired when tumor cells respond to anticancer therapy, but, during cell proliferation, form therapy- or drug-tolerant cells with gene mutations or changes in protein expression [3,24]. Progenitor cells, known as stem cells, can proliferate longer than non-stem cells and have the properties to create multiple cell types throughout the organism. These abilities indicate that progenitor cells are not differentiated [27,28,29]. However, cancer progenitor cells have proliferation potential. Normal stem cells are normally in a resting or inactive quiescent state (in G1/G0) and are preserved from cell damage or mutation [27].

Cancer progenitor cells (CPCs) have the characteristics of stem cell-like cells and are considered to be the main cause of tumor heterogeneity because they are capable of producing the entire repertoire of cancer cell types [2,5,30]. In addition, CPCs can form tumors; in contrast, non-tumorigenic cancer cells cannot form. DTP (drug-tolerant persister) cancer cells have been found in colorectal cancer cells, basal breast cancer cells, and non-Hodgkin lymphoma cells [31,32,33]. DTP cells reduce growth and alter metabolism, facilitating greater tolerance to drugs. The population of DTP cells is small, and the cells arise with a low incidence of about 0.3% to 5% [31,34,35]. According to the literature, micronuclei, which are intracellular particles formed by DNA damage or Howell–Jolly bodies, appear in blood smears in patients with deficient spleen function [36]. There are no specific markers for Howell–Jolly bodies. Howell–Jolly bodies are associated with the possibility of developing cancer; micronuclei are parts that contain the DNA of damaged cells, which is not the same as microcells [36]. Buiķis and colleagues have observed microcells that are small, rounded, and spindle-shaped. Microcells were intensively stained with azure eosin dye [37]. The microcell population has been observed to increase after 24 to 48 h in a human sarcoma cell line HT-1080, a Dzungarian fibroblastoma cell line, and a HeLa human cervical cancer cell line [37,38,39]. The microcell population increases to 1% as a result of treatment [38,40]. Resistance to DTP cells may be reversible after treatment if proliferation is low [35]. However, DTP cells have rare mutations that occur in a dormant state, and these cells can proliferate in the presence of anticancer agents [32]. Although microcells form, applied therapy still has an effect, and due to cultivation, form and size changes are observed. Microcells develop through sporosis. This is a process of microcell formation from a damaged tumor microcell, in which the perinuclear bodies or nucleoli of the damaged macrocell are removed [38]. Microcells have endocytic ability, previously demonstrated in NRU (neutral red uptake) assays by active transport, which means that these cells are viable [40]. Microcells have also shown accumulation [39], in a study of colorectal cancer, of small and large cell populations of 10 µm and 25 µm, respectively. The small cell population had greater invasion and metastatic capacity than large cells [41].

Each cell in an organism has multiple characteristics, and these cells can be identified by their markers. Cell markers are variable and different, such as cluster differentiation number (CD), cell surface markers, and markers to identify SCs, CSCs, and metabolic active cells [42,43]. Markers that indicate stemness are SOX2 and Nanog. SOX2 is involved in stem cell maintenance during embryogenesis. It plays an important role in the cell regeneration process, reprogramming and homeostasis, and promotes proliferation and cell survival [8]. CD24 and CD44 are used as breast cancer stem cell markers [9,30,44]. DTP cancer cells and small cells have stem cell-like characteristics [24,41,45]. These cells express SOX2, Nanog, and Oct3/4, embryonic stem cell markers. The aldehyde dehydrogenase (ALDH) protein functions as an antioxidant, and the CD44 protein is associated with migration and potential invasiveness [24,33,41,45,46].

In this study, the size of the microcells and the marker expressed were determined, as well as whether the microcells synthesized RNA, indicating the ability of the cells to regenerate and repair their genes. Sk-Mel-28 melanomas treated with PTX were examined. Small cells were identified in aggressive cancer lines, LoVo and HT-29 colorectal cancer cell lines, as well as the HeLa cell line for human cervical cancer. Therefore, this study aimed to examine melanoma microcells morphologically and immunocytochemically.

## 2. Materials and Methods

### 2.1. Cell Lines

Melanoma Sk-Mel-28 cells (ATCC^®^ HTB-72™) and human lung carcinoma A549 cells (CRM-CCL-185™) cells were obtained from the American Type Culture Collection (ATCC, Manassas, VA, USA). Sk-Mel-28 cell line expresses mutant B-Raf (V600E) and wild-type N-Ras. It also expresses antigens as well as blood type A; Rh+; and HLA A11, A26, B40, and DRw4. Cells were cultured and maintained in Dulbecco’s modified Eagle medium (DMEM; 41965120 or 21063029, without phenol red; Thermo Scientific, Waltham, MA, USA) with 10% fetal bovine serum (FBS; F7524, Sigma-Aldrich, Saint Louis, MO, USA), 1% penicillin/streptomycin (30-002-CI, Corning^®^, Corning, NY, USA) at 37 °C in a humidified atmosphere of 95% CO_2_. Cells were seeded at a density of ~1 × 10^5^ cells/mL and examined at ~80% confluence.

### 2.2. Cell Treatment

The day before treatment, Sk-Mel-28 cells were seeded on round coverslips on a 24-well plate at a density of ~1 × 10^5^ cells/well to reach 80% confluence. During the experiment, cells were treated with paclitaxel, while control cells did not receive treatment. Paclitaxel (PTX; 6 mg/mL; TEVA, Accord healthcare B.V., Utrecht, The Netherlands) was added at a final concentration of 0.7 µM in culture medium (DMEM with 10% FBS and 1% penicillin/streptomycin) and incubated for 24 h at 37 °C in a humidified atmosphere of 95% CO_2_. The medium was changed and rinsed twice with 1X phosphate-buffered solution (PBS), added to the fresh culture medium, and incubated for 24 h at 37 °C in a humidified atmosphere of 95% humidified 5% CO_2_. At the end of the experiment, the cultivation medium was removed, and fixation with a 4% formaldehyde solution (37% formaldehyde diluted in PBS to obtain a 4% formaldehyde solution; 8.18708; Sigma-Aldrich, Saint Louis, MO, USA) was carried out for 10 min at room temperature, followed by rinsing twice with 1X PBS for 5 min, before moving on to immunocytochemistry.

### 2.3. Immunocytochemistry

The immunocytochemistry method was used to detect cells expressing proteins. Cells were grown on coverslips and fixed with a 4% formaldehyde solution for 10 min at room temperature and then washed twice in 1X PBS for 5 min. Subsequently, the slides were permeabilized for 10 min in 0.02% Triton X-100 (X100; Sigma-Aldrich, Saint Luis, MO, USA) and rinsed three times in 1X PBS for 5 min. For slide blocking, 2% bovine serum albumin (BSA)/Tris-buffered saline containing 0.2% Tween 20 (E108; Nordic BioSite, Tampere, Finland; TBST) was used for 1 h at room temperature. Cells were incubated overnight with a monoclonal antibody, according to the manufacturer’s recommendation, in a humidified chamber at +4 °C. Sequentially, the incubation solution was decanted from the samples, and the samples were not washed. Subsequently, the samples were covered with secondary antibody diluted in 1%BSA/TBS and incubated in the dark for 1 h at room temperature in a humidified chamber. For staining sequences with multiple antigens, blocking was performed after incubation with the secondary antibody; subsequently, primary and secondary antibodies were added. Cell nuclei were counterstained with 1 µg/mL DAPI (02157574-CF; MP Biomedicals, Solon, OH, USA) for 1 min and embedded in CV Ultra Mounting Medium (14070936261; Leica Biosystems, Nussloch GmbH, D Nussloch, Germany). Expression was examined by two independent observers as a negative control and positive staining. Primary and secondary antibodies are listed in Table 1.

### 2.4. RNA Synthesis Detection

An RNA synthesis assay kit (ab228561, Abcam, Cambridge, UK) was used for the detection of RNA synthesis. Sk-Mel-28 cells were seeded in a 24-well plate at a density of 1 × 10^5^ cells/mL and cultured in DMEM with 10% FBS and 1% penicillin/streptomycin at 37 °C in a humidified atmosphere with 5% CO_2_. The cells were grown on coverslips. When cells reached 80% confluence, two groups were distributed as control and PTX samples. For the control samples, the cultivation medium was changed. Conversely, for PTX samples, the cultivation medium was changed and PTX was added at a final concentration of 0.7 µM; the cultivation continued for 24 h. A 1X RNA label was added to the control and PTX cells. After incubation for 24 h, cells were fixed and processed according to the manufacturer’s instructions [47]. Cells were analyzed by microscopy.

### 2.5. DNA Synthesis Detection

DNA synthesis detection was used to assess the ability to proliferate after treatment with anticancer drugs. DNA synthesis detection was performed with Click-iT^®^ EdU flow cytometry assay kits (Cat.no. C10419, Molecular Probes by Life Technologies, Eugene, OR, USA), that is, the BrdU (bromo-deoxyuridine) assay analog. 5-ethynyl-2-deoxyuridine (EdU) is a nucleoside analog of thymidine and is involved in DNA during active DNA synthesis. EdU is in the ethynyl moiety, and the visualization agent is azide-coupled to Alexa Fluor^®^ 647 dye to enable detection by flow cytometer. Sk-Mel-28 cells were seeded in a 24-well plate at a density of 1 × 10^5^ cells/mL and grown in DMEM with 10% FBS and 1% penicillin/streptomycin at 37 °C in a humidified atmosphere with 5% CO_2_. When the cells reached 80% confluence, two groups were created as control and PTX samples, and subgroups—control with and without EdU and PTX group—with and without EdU. For the control samples, cultivation medium was added. For the PTX samples, the cultivation medium was changed and PTX was added at a final concentration of 0.7 µM. Both groups were cultivated for 24 h. Sequentially, the medium was replaced with EdU culture medium with EdU at a final concentration of 10 mM, and the cultures were kept for another 24 h. The cells were then trypsinized and fixed in 4% formaldehyde. Incorporation of EdU was determined with the Click-iT EdU Alexa Fluor 647 flow cytometry assay kit (Molecular Probes) according to the manufacturer’s instructions.

### 2.6. Flow Cytometry and Data Analysis

Flow cytometry was performed using a three-laser BD Aria II flow cytometer (BD—Becton Dickinson, Biosciences, San Jose, CA, USA). The analysis was carried out with Flowing Software 2.5.1. and IDEAS 6.2. software. Singlets were analyzed by size using an FCS scatter plot (forward scatter) vs. SSC (side scatter) to determine the small cell population; all cells and populations by size were compared using PI (Propidium Iodide; PI flow cytometry kit 200 test, Ex.493/Em.636 nm, Cat.no ab139418, Abcam) and EdU (Click-iT EdU Alexa fluor 647 flow cytometry assay kit, 100assays, Cat. No C10419, Ex.650/Em.670 nm flow cytometer laser lines, Molecular Probes Life Technologies) median fluorescence intensity. The cell cycle phases were determined by the intensity of the PI stain. EdU fluorescence was determined using an APC laser.

### 2.7. Microscopy

A Leica SP8 confocal microscope (Leica Microsystems, Wetzlar, Germany) with a 63× objective (oil, planapochromatic, with a numeric aperture of 1.40; Leica Microsystems, Wetzlar, Germany) was used. A ZEISS Axiolab 5 microscope with a 63×/0.85 objective (oil, N-Achroplan, Ph3 M27; ZEISS, Jena, Germany) and a ZEISS Axiocam 202 monomicroscopy camera (ZEISS, Jena, Germany) were used for the microscopy of the samples. For image analysis and processing, Image-Pro^®^ Plus, version 4.0, LAS X lite (Leica Microsystems, Wetzlar, Germany) and ZEN 3.0 Blue lite (ZEISS, Jena, Germany) were used.

## 3. Results

Melanoma cells (Sk-Mel-28) were used in our investigation to characterize microcells that formed after immunocytochemical treatment with chemotherapeutic drugs. An experiment was conducted on human lung carcinoma A549 cells (CRM-CCL-185^TM^) treated with PTX, and after 48 h, the cells were processed with a PCNA antibody to analyze antigen expression. Microcells are small round cells whose formation inhibits anticancer drugs such as thiophosphamide (Thio-TEPA) [37]. Our study used PTX, a microtubule inhibitor that stops mitotic division to induce microcells [18].

### 3.1. Caspase-2 Expression after Paclitaxel Treatment in Microcells

Caspase-2 (Casp-2) is an initiator caspase belonging to the caspase family of proteases, which stimulates the initiation of the apoptotic cascade [48,49]. Casp-2 was expressed in the cell nucleus in the control sample (Figure 1A) with green fluorescence. Cell nuclei were counterstained with DAPI in fluorescent blue (Figure 1). Casp-2 expression was equal and smooth in the nucleus of the control cell, while in treated cells, the expression was different (Figure 1B). The expression of Casp-2 in the nucleus of the cell was not smooth and was stronger in large polyploid cells with multiple nuclei (Figure 1BI,BII). The nuclei were of different sizes and contained various amounts of DNA that was shown by staining with DAPI (Figure 1BI,II). Microcell nuclei were 2.5, 3.37, and 4.61 µm in diameter (Figure 1BI,II). The microcell nuclei strongly expressed Casp-2 (Figure 1BI,II), and the amount of DNA in the microcell was high, as shown by DAPI.

### 3.2. Caspase-6 Expression after Paclitaxel Treatment in Microcells

During apoptosis, Caspase-6 (Casp-6) is an effector caspase similar to Caspase-3 and is generally cleaved and activated by an initiator caspase such as Casp-2 [50]. Casp-6 was found in the cytoplasm of Sk-Mel-28 control cells, but was more abundant in the cell periphery and cell membrane (Figure 2A). In turn, Casp-6 from Sk-Mel-28 cells treated with PTX was expressed in cell nuclei (Figure 2B) and microcells (Figure 2B, red arrowhead). The nuclei were counterstained with DAPI (blue).

### 3.3. Expression of Aldehyde Dehydrogenase and Caspase-3 after Paclitaxel Treatment in Microcells

ALDH2 (aldehyde dehydrogenase) is an enzyme that characterizes cell metabolic activity [42], indicating the ability of cells to resist treatment. ALDH2 is expressed in the cytoplasm of control and PTX-treated melanoma cells (Figure 3A,B, red). However, in cells treated with PTX, the expression of ALDH2 is stronger.

Caspase-3 (Casp-3) is involved in the execution of apoptosis and plays a role in the evaluation of tumor regression [51]. Casp-3 expression was not observed in control cells (Figure 3A, green), while expression in cells was observed in the cell cytoplasm (Figure 3B, green).

### 3.4. SOX2 and Nanog Expression after Paclitaxel Treatment in Microcells

SOX2 and Nanog are stem cell markers associated with pluripotency, cell regulation, and reprogramming [52,53]. SOX2 was expressed in the cell nuclei (Figure 4A, green) and Nanog was expressed in the cell cytoplasm (Figure 4A, purple) in control cells. SOX2 was strongly expressed in cell nuclei but weakly in the cytoplasm in cells (Figure 4B, green), while Nanog was expressed in the cytoplasm (Figure 4B, purple). However, in PTX-induced microcells (Figure 4B, white arrows), SOX2 and Nanog were observed in the cell region between small nuclei.

### 3.5. Nuclear Antigen Expression after Paclitaxel Treatment in Microcells

Nuclear proliferation cell antigen (PCNA) plays an essential role in normal DNA synthesis and replication [54]. This experiment was carried out in two different cell lines: melanoma (Sk-Mel-28) and human lung carcinoma (A549) to show that microcells are not specific to any particular cancer cell line. Microcells are observed in HeLa and the human sarcoma cell line HT-1080 [37,38,39]. PCNA expression was observed in the cell nuclei in Sk-M-28 control cells (Figure 5A, control). However, this expression was weak and was also observed in the cell cytoplasm. In PTX treated with PTX, PCNA was expressed only in cell nuclei (Figure 5B, PTX treated). The expression of PCNA was not equal in all nuclei. There were round nuclei where the expression of PCNA was moderately strong compared to that of multinuclear cells. In multinuclear cells, there were a few microcell nuclei that strongly expressed PCNA (Figure 5B, PTX treated, red square, white star). The microcells could be separated by the PCNA expression region. PCNA was expressed in all cells, but not in all microcells. PCNA was expressed in the periphery of the nucleus, nucleus, and cell cytoplasm of microcells. A549 control cells (Figure 5C, control A549) showed different expressions of PCNA. Expression was evaluated in the nuclei of the cell, but some cells showed expression in the periphery of the nucleus. Microcell formation (Figure 5D, A549) was observed in the A549 cell line after PTX treatment. The PCNA expression antigen was observed in microcells in the periphery of the nucleus (Figure 5D, A549 treated with PTX, white arrow). The expression of PCNA in both the melanoma line and the A549 line of microcell subpopulations shows that DNA repair-related processes occurred. PCNA expression was mainly related to cell replication and was involved in DNA repair systems; otherwise, in control cells, PCNA expression was observed in the nuclei and cytoplasm [53,55,56,57]. It played an important role in cell cycle regulation, especially with high expression during the G1 and S phases [58].

### 3.6. Ability of Microcells to Regeneration of DNA

Previous experiments have shown the formation of microcells and cells that are intensely stained with basic dyes [38,54]. Propidium iodide (PI) dye was used to assess the distribution of DNA abundance in the Sk-Mel-28 cell population under the influence of PTX. Figure 6 shows the distribution of cellular DNA on a histogram. In the control sample, a clustering distribution is observed around the DNA diploid region (Figure 6A). After 48 h under the influence of PTX, both cells with a higher amount of DNA and small cells with a decreased amount of DNA are observed in the Sk-Mel-28 cell population (Figure 6B). The results of subsequent studies using the EdU marker confirmed the viability of microcells in the small cell population.

Microcell proliferation was examined using flow cytometry. The EdU substrate was added to both Sk-Mel-28 control cells and PTX-treated cells. EdU is a thymidine nucleoside analog and is activated during DNA synthesis [59]. When analyzing the dot plot of the flow cytometry data, a small region of cells can be distinguished with high levels of DNA synthesis (Figure 7). Microcells can be seen to have a high level of DNA synthesis. The number of microcells in the control group was 0.84% and increased to 4.42% with PTX treatment (Table 2).

Furthermore, the ratio of DNA synthesis to the amount of nucleic acids (EdU/PI) of positive microcells is significantly higher than that of the cell population (Table 2).

The ratio indicates an increased accumulation of DNA in the microcell compared to the rest of the population. They are metabolically more active cells with high proliferative potential.

### 3.7. RNA Synthesis in Microcells after Paclitaxel Treatment

During RNA labeling, it is possible to detect newly synthesized RNA [47]. The de novo synthesized RNA demonstrates the ability to survive after PTX treatment, as RNA is crucial for gene translation and protein expression in living cells. RNA is observed in all control cell nuclei (Figure 8, control). However, RNA synthesis is detected in various cell nuclei in cells (Figure 8, PTX-treated). RNA synthesis is not equal in all cells; it is observed in polyploid cell nuclei (Figure 8, PTX treated I), microcells (Figure 8, PTX treated II), and macrocell nuclei (Figure 8, PTX treated III). It appears that there are defined microcell subpopulations that are characterized by RNA+ and RNA synthesis. Macrocells that developed after PTX treatment were also characterized by two subpopulations: RNA+ and RNA synthesis.

In summary of these results, microcells are characterized by the expression of Casp-2, Casp-6, Casp-3, ALDH2, SOX2, Nanog, PCNA, and RNA synthesis in cell nuclei, as well as in cell cytoplasm.

This section can be divided into subheadings. It should provide a concise and precise description of the experimental results, their interpretation, and the experimental conclusions that can be drawn.

## 4. Discussion

Microcells are described as a natural tumor component, and their count increases after cancer therapy. They are small cells with a thin layer of cytoplasm and can be intensively stained with basic dyes [37]. Sporosis is the process of their formation from damaged tumor macrocells. After anticancer chemotherapy in tumor tissues, the number of microcells increases rapidly [37,40,54]. In previous research, we reported that the number of microcells increased 48 h after anticancer therapy in Sk-Mel-28 and HeLa cells; in the non-cancerous cell line HS-68 (human fibroblasts), microcells were not observed in control but in treated cell samples, and 1 microcell was observed out of 1060 total counted cells [40]. Microcells are related to resistance to tumor therapy or tolerance to cancer drugs. These microcells are characterized by antigens and markers. Markers include molecules such as proteins, nucleic acids, antibodies, and peptides, among others [53,54,60,61].

It is well known that apoptosis is induced by chemotherapy acting on tumors [62,63]. Apoptosis results in nucleolar condensation and breakdown. Multiple caspases characterize the steps of initiation, activation, and execution of apoptosis [64,65,66]. The most researched is Casp-3, which is activated and performs proteolysis. However, research shows that Casp-3 activates cell growth and proliferation without causing cell death [62,67]. Our study shows similar observations. Casp-3 expression is observed in melanoma cells after PTX treatment compared to control cells. ALDH2 expression is also detected in both control and PTX-treated cells. Aldehyde dehydrogenase 2 (ALDH2) is a member of the ALDH family of proteins and is an enzyme that oxidizes alcohol. ALDH2 expression is related to high metabolic activity in a cell through the cytosol or mitochondria, distinguished by its electrophoretic motilities. PTX-treated cells show stronger ALDH2 expression in the cell cytoplasm than control cells. ALDH2 metabolizes alcohol derivatives to promote aldehyde dehydrogenation [11]. Cancer cells that have been shown to express ALDH are resistant to chemotherapy, e.g., paclitaxel, doxorubicin, and gemcitabine. The presence of ALDH in cells is also associated with tumor precursor cells [9,11]. In turn, Casp-2 belongs to the caspase family [68]. Casp-2 is normally expressed in the cell nucleus, but when cells are treated with the microtubule inactivator vincristine, expression is observed in the cell cytoplasm [69]. We observed Casp-2 expression in all control cell nuclei; PTX-treated cells showed weaker expression in nuclei but strong expression in microcells. The strong expression of Casp-2 in the nuclei indicates the ability to evade anticancer treatments [69,70]. In the current study, strong expression of Casp-6 was also detected in microcells that form after PTX treatment nuclei. However, in control cells, Casp-6 is expressed in the cell cytoplasm in the nucleus area. According to the literature, Casp-6 expression is associated with nuclear laminas in cancer cells, characterized by tumor initiation [71]. Caspase research studies indicate that Casp-3 could mediate proangiogenic effects in dying tumor cells [66].

The markers that indicate stemness are SOX2 and Nanog. SOX2 is involved in stem cell maintenance during embryogenesis. It plays an important role in the cell regeneration process, reprogramming, and homeostasis, and promotes proliferation and cell survival [8]. SOX2 expression is observed early in embryonic development [72,73]. SOX proteins are determined mainly as DNA binding elements. They are called high mobility groups (HMGs) and are associated with mature transcription regulators (i.e., SEX determining factors Y and SRY) so that SOX/Sox proteins are functionally DNA binding elements. They play an important role in various stages of embryonic development and preservation of independent embryonic stem cells (EMCs) [74,75]. Several researchers have observed a connection between SOX2 expression and various clinical aggressions, as well as the development of cancer resistance, including lung, breast, and prostate cancers [75]. Overexpression and amplification of SOX2 have been observed in many types of tumors, leading to accelerated migration, proliferation, invasion, and metastasis of cancer cells, ultimately resulting in resistance to apoptosis [8]. In our investigation, we observed SOX2 expression in control cell nuclei, while cytoplasmic expression and expression in microcells in cells treated with PTX were identified. The expression of SOX2 in the cytoplasm indicates the aggression of cancer cells and the ability to survive applied stress [8,76]. Nanog expression was observed in the cell cytoplasm of control and PTX treated cells, as well as microcells. Nanog expression has been observed in cancers with a poor prognosis [8,76]. To date, studies have shown that cancer stem cells have DNA repair mechanisms, avoiding the effects of drugs [9,77]. Cell proliferation and protein synthesis are characterized by PCNA and RNA synthesis [47,53,78]. In control cells, PCNA expression was observed in the cell nucleus–cytoplasm region. After PTX treatment, we observed that the microcell nuclei expressed PCNA, indicating that the microcells contained DNA. The increase in PCNA expression represents a high replication activity in microcells and indicates their high capacity for renewal. PCNA expression is observed in cancer cell nuclei, not in normal cells [58,79]. In some studies, PCNA expression was observed in the cancer cell cytoplasm not in normal cells and is related to cell stress [79,80]. Strong expression in cancer cell nuclei is significantly associated with malignant tumors, and high expression has been associated with pathological grade, stage of TNM (tumor, node, and metastasis) and the number of metastases [58]. RNA synthesis is observed in all control cell nuclei, but not in PTX-treated cells. PTX-treated cells could be divided into subpopulations according to RNA synthesis. There are microcells and macrocells with RNA synthesis, and microcell subpopulations without RNA synthesis. This indicates cell activity and the ability to react or avoid cancer treatment. Microcells that have detected PCNA expression and RNA synthesis can tolerate applied therapy with cancer stem cell properties. There was a propensity for large cells with low DNA concentration and tiny cells with low DNA content to predominate in the luminal population of breast cancer cells. Conversely, a population of triple-negative breast cancer cells included large and small cells, as well as cells, with high and low DNA concentrations. Flow cytometry data show that PTX-treated melanoma cells are small with a high EdU content, indicating DNA synthesis (Figure 6 and Figure 7). DNA synthesis is indicative of the ability of a cell to develop and proliferate [55,81]. Microcells can actively proliferate, as shown by the incorporation of EdU into DNA in the active synthesis phase, as the well as RNA synthesis detected by the RNA synthesis assay.

PTX-treated cancer cells show stronger ALDH2 than untreated cells. This enzyme demonstrates cell self-defense and resistance to the use of alkylating agents in anticancer therapy. Microcells form during the application of the anticancer drug PTX. Microcells have a higher renewal capacity and the ability to differentiate into pluripotent progeny cells indicating an undifferentiated cell. Although microcells express apoptosis markers that indicate programmed cell death, these markers can also indicate the cell’s ability to tolerate the therapies. The synthesis of ALDH, PCNA, and RNA indicates the metabolic activity of the cells, which promotes resistance to applied therapy. Consequently, it is important to continue research and develop individualized microcell formation detection methods after anticancer therapy, thus determining the effectiveness of the applied treatment.

## 5. Conclusions

This morphological study reveals that microcells with extremely high levels of DNA synthesis can contribute to resistance. It appears that the progenitors of the resistant cell population may be microcells. After taking the medication, the number of microcells increased. These cells are also detected in small numbers in the control material of malignant tumors, suggesting that they are a subpopulation of natural cancer cells. Therefore, the analysis of microcell populations in tumors receiving anticancer therapy can be an important predictor of patient survival and a possible source of cancer cell regeneration after tumor death. Stopping the development of microcells would contribute to overcoming tumor resistance and significantly improve conventional anticancer therapy.

## Figures and Tables

**Figure 1 biomedicines-12-01576-f001:**
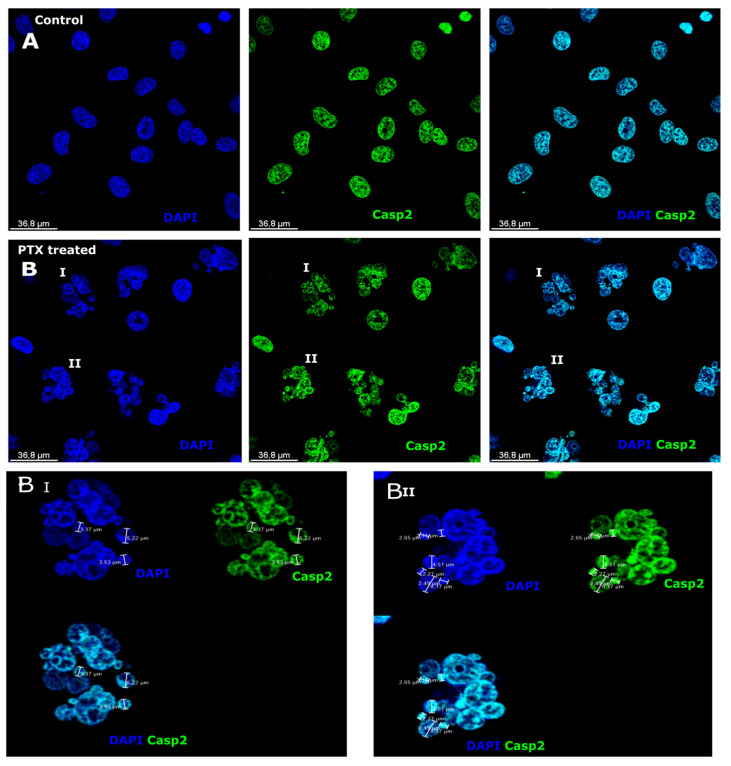
Caspase-2 expression in PTX-induced microcells. (**A**) Untreated Sk-Mel-28 cells (control cells). Cell nuclei are counterstained with DAPI (blue). Caspase-2 (green) is expressed in the cell nuclei. (**B**) Sk-Mel-28 cells. Cell nuclei are counterstained with DAPI (blue); Caspase-2 (green) is expressed in cell nuclei, although stronger expression of Caspase-2 is evaluated in microcells (**BI**,**BII**). Micro- and macrocells I and II treated with PTX are enlarged to show the PTX-treated (**BI**) and (**BII**) macrocells and adjacent microcells.

**Figure 2 biomedicines-12-01576-f002:**
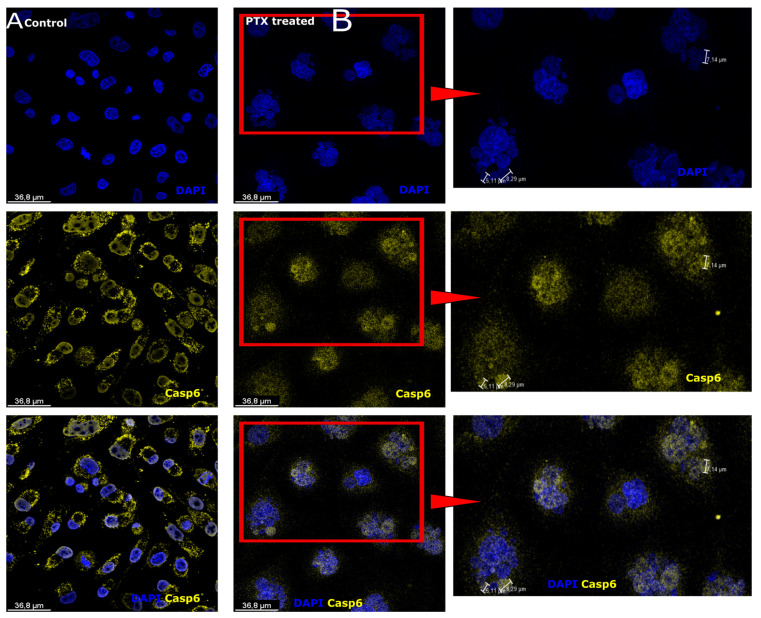
Caspase-6 expression in microcells. (**A**) Sk-Mel-28 cells expressed Caspase-6 (Casp-6; yellow) in the cell cytoplasm; cell nuclei are counterstained with DAPI (blue). (**B**) Sk-Mel-28 cells expressed Caspase-6 (Casp-6; yellow) in the nucleus area and microcells (red arrowhead); Cell nuclei are counterstained with DAPI (blue), the region of interest (red square) enlarged on the right side.

**Figure 3 biomedicines-12-01576-f003:**
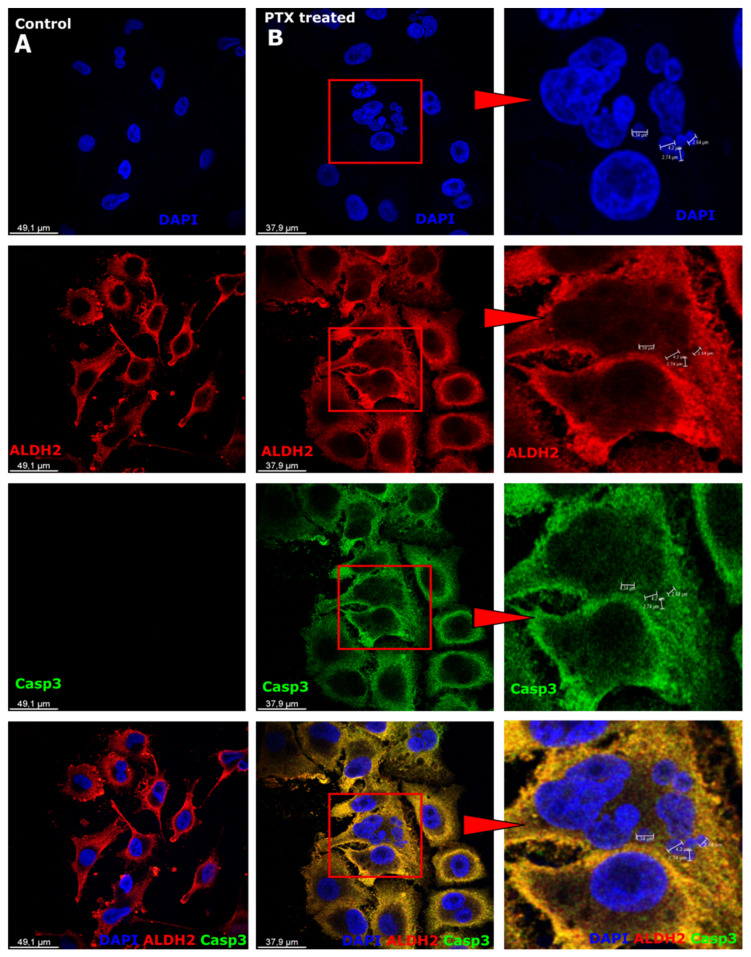
Expression of aldehyde dehydrogenase and caspase-3 in melanoma cells. (**A**) Sk-Mel-28 cells expressed aldehyde dehydrogenase (ALDH2; red) in the cell cytoplasm; caspase-3 (Casp-3; green) expressed in the cell cytoplasm; cell nuclei are counterstained with DAPI (blue). (**B**) Sk-Mel-28 cells with aldehyde dehydrogenase (ALDH2; red) in the cell cytoplasm; caspase-3 (Casp-3; green) expressed in the cell cytoplasm; cell nuclei are counterstained with DAPI (blue), the region of interest (red square) enlarged on the right side.

**Figure 4 biomedicines-12-01576-f004:**
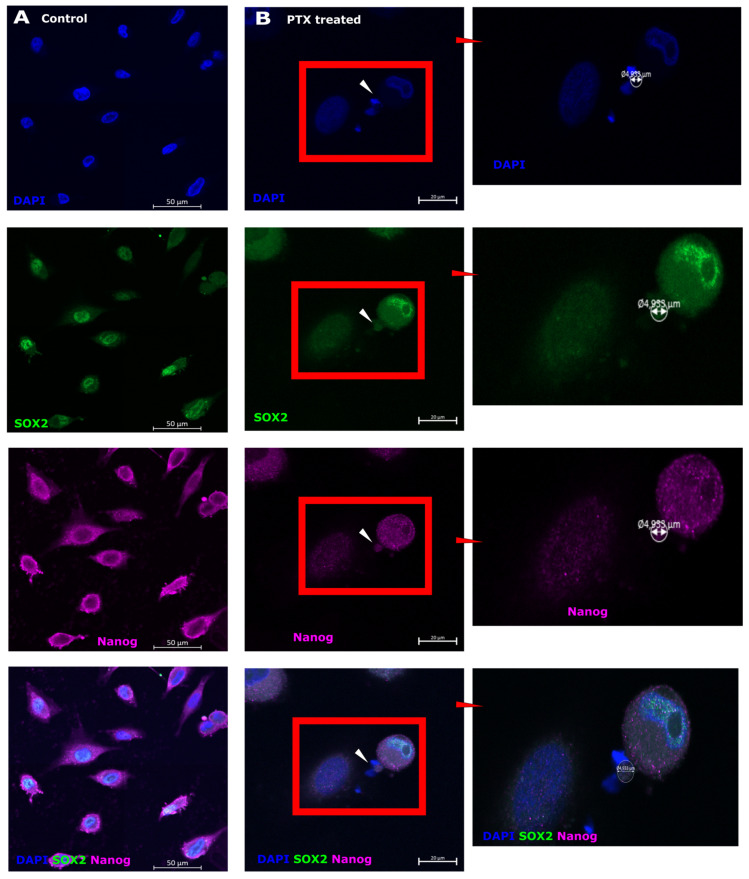
Expression of SOX2 and Nanog in microcells. (**A**) Sk-Mel-28 cells expressed SOX2 (green) in cell nuclei; Nanog (purple) is expressed in the cell cytoplasm; cell nuclei are counterstained with DAPI (blue). (**B**) Sk-Mel-28 cells express SOX2 (green) in the nuclei and cytoplasm; Nanog (purple) is expressed in the cell cytoplasm; cell nuclei are counterstained with DAPI (blue). Microcells (white arrows) express both SOX2 and Nanog; the region of interest (red square) is enlarged on the right side.

**Figure 5 biomedicines-12-01576-f005:**
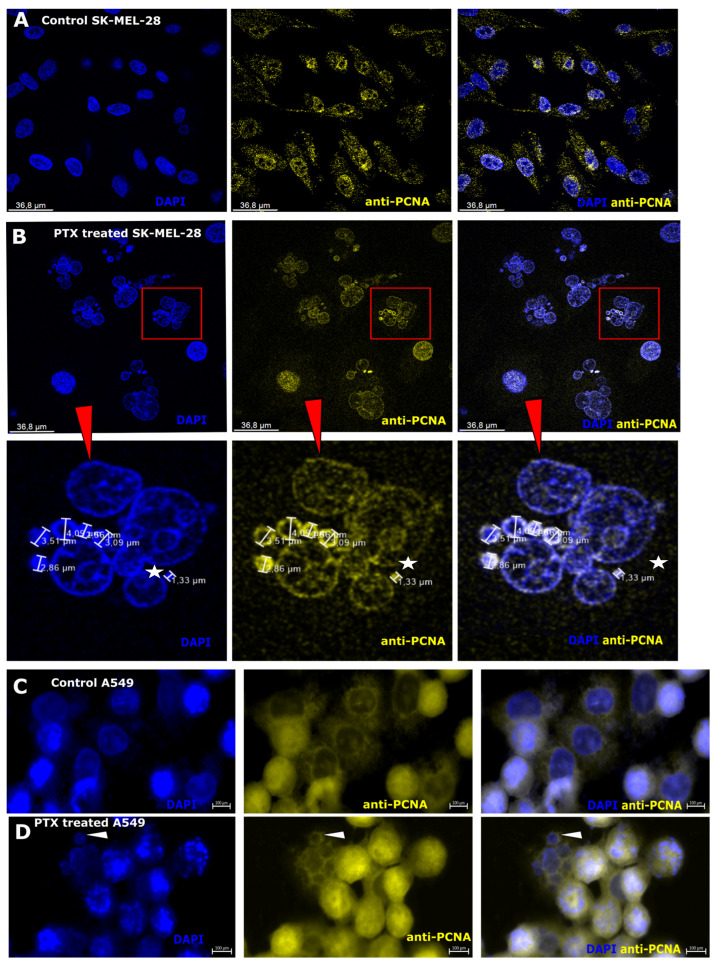
Anti-PCNA expression in melanoma and human lung carcinoma cells. (**A**) Sk-Mel-28 cells weakly express anti-PCNA (yellow) in the cell nuclei; cell nuclei are counterstained with DAPI (blue). (**B**) Sk-Mel-28 cells express anti-PCNA (yellow) in the cell nuclei and strongly express it in microcells; cell nuclei are counterstained with DAPI (blue). Microcells (white star) strongly express anti-PCNA (yellow); region of interest (ROI; red square) is enlarged below. (**C**) Control, A549 cells express anti-PCNA (yellow): The parts of the cells show strong expression, while the other parts of the cells show weak expression; cell nuclei are counterstained with DAPI (blue). (**D**) A549 cells express anti-PCNA (yellow) in cell nuclei and microcells; cell nuclei are counterstained with DAPI (blue). The microcells (white arrowhead) express anti-PCNA (yellow) in the periphery.

**Figure 6 biomedicines-12-01576-f006:**
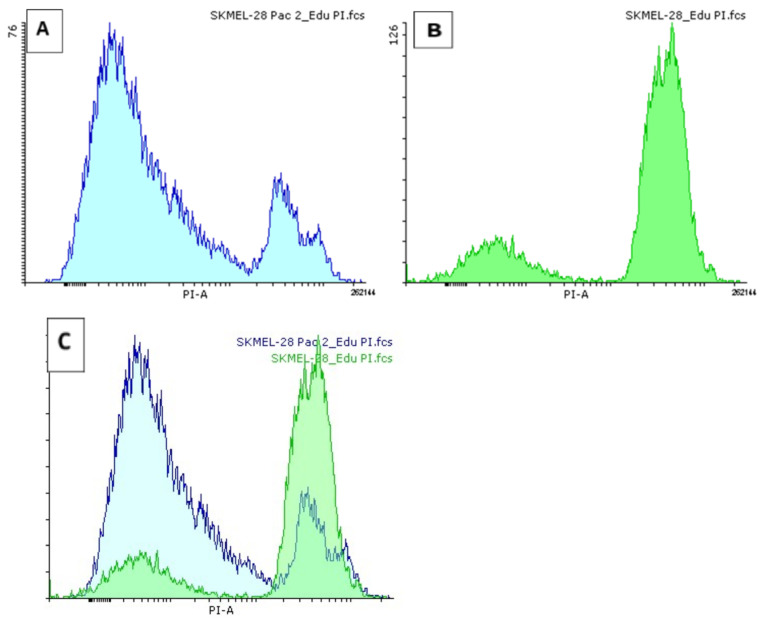
DNA distribution in the melanoma cell line 48 h after PTX treatment. (**A**) Sk-Mel-28 cell control sample, PI-stained nucleic acids (propidium iodide); (**B**) Sk-Mel-28 cells treated with PTX and PI-stained nucleic acids stained with PI. (**C**) Overlay of (**A**,**B**).

**Figure 7 biomedicines-12-01576-f007:**
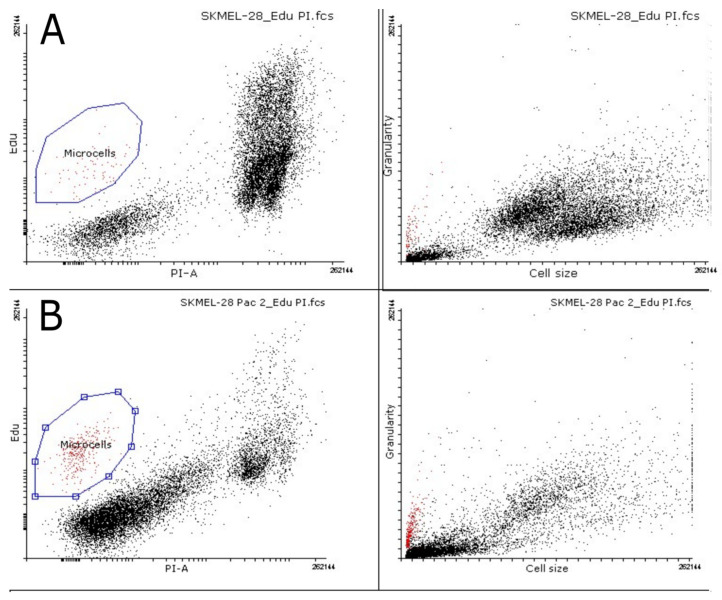
Flow cytometry data showing DNA synthesis in melanoma cells after PTX treatment. (**A**) Sk-Mel-28 control cells, stained with propidium iodide and marked EdU. (**B**) Sk-Mel-28 PTX treated cells, stained with propidium iodide, and marked EdU.

**Figure 8 biomedicines-12-01576-f008:**
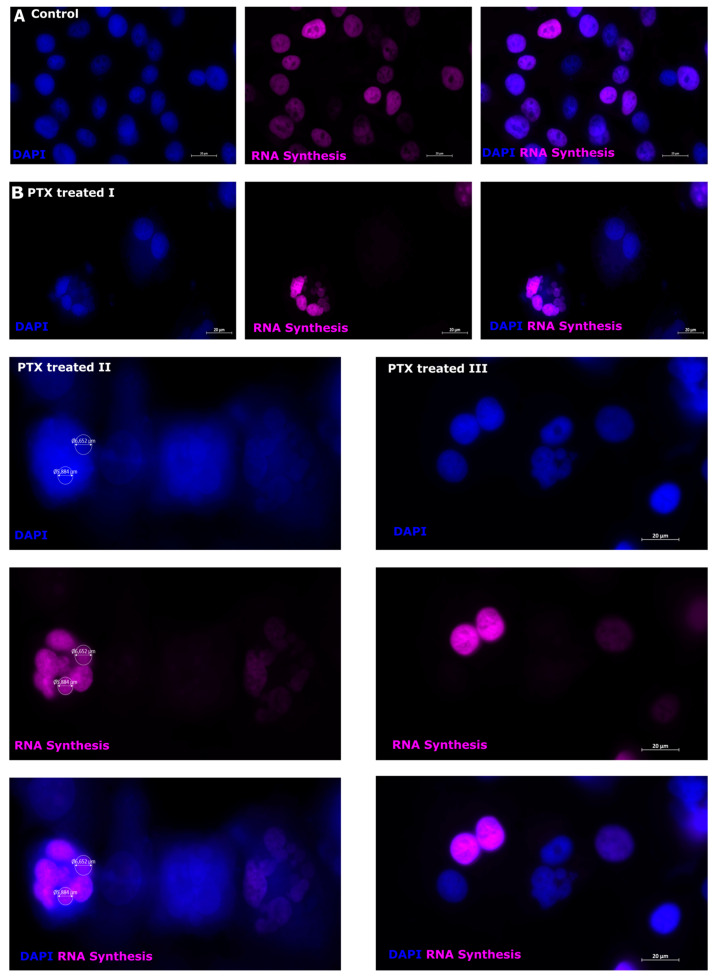
RNA synthesis in cells treated with PTX. (**A**) Sk-Mel-28 cells show RNA synthesis labeling (purple) in all cell nuclei; cell nuclei are counterstained with DAPI (blue). (**B**) Sk-Mel-28 treated cells indicate RNA synthesis (purple) in polyploid cell nuclei (I), microcells (II), and macrocell nuclei (III); cell nuclei are counterstained with DAPI (blue).

**Table 1 biomedicines-12-01576-t001:** Primary and secondary antibodies.

Antibody	Description	Antibody Emission and Excitation	Dilution/Used Concentration	Product No and Manufacturer
**ALDH2**	Primary antibody: rabbit anti-human Rabbit polyclonal		1:75	158-201549-T08-200, Sino Biological by Nordic BioSite, Täby, Sweden
**SOX2**	Primary antibody: mouse anti-human antibody Mouse monoclonal		1:100	TA302025, Origene, Rockville, MD, USA
**Anti-PCNA**	Primary antibody: rabbit anti-human, rabbit recombinant monoclonal		1:100	ab92552, Abcam, Amsterdam, The Netherlands
**Nanog**	Primary antibody: rabbit anti-human, rabbit polyclonal		1:100	14295-1-AP, Proteintech Europe, Manchester, UK
**Caspase-2**	Primary antibody: rabbit anti-human, rabbit polyclonal	FITC Conjugated; Ex.494 nm/Em.518 nm	1:100	BS-5802R-FITC, Bioss antibodies/Nordic BioSite, Täby, Sweden
**Caspase-6**	Primary antibody: rabbit anti-human, rabbit polyclonal	ALEXA FLUOR^®^ 555 conjugated Ex.553 nm/Em.568 nm	1:100	BS-0151R-A555, Bioss antibodies/Nordic BioSite, Täby, Sweden
**Caspase-3**	Primary antibody: mouse anti-human, mouse monoclonal		1:100	GTX13586, Biolegend/GeneTex Inc., Tampere, Finland
**Alexa fluor 488**	Secondary antibody: goat anti-rabbit Igg H&L	Ex.495 nm/Em.519 nm	1:1000	ab150077 Abcam, Carlsbad, CA, USA
	Secondary antibody: goat anti-mouse Igg H&L	Ex.495 nm/Em.519 nm	1:1000	ab150113 Abcam, Carlsbad, CA, USA
**Alexa fluor 594**	Secondary antibody: goat anti-mouse Igg H&L	Ex.590 nm/Em.617 nm	1:1000	ab150116 Abcam, Carlsbad, CA, USA
	Secondary antibody: goat anti-rabbit Igg H&L	Ex.590 nm/Em.617 nm	1:1000	ab150088 Abcam, Carlsbad, CA, USA

**Table 2 biomedicines-12-01576-t002:** Cell fluorescence intensity ratio EdU/PI.

Microcells		
	Microcell Count	Mean (EdU/PI)
**Sk-Mel-28_EdU/PI (control)**	84	20.6
**Sk-Mel-28 PTX 2_EdU/PI.**	442	18.7
**All cells**		
**Sk-Mel-28_EdU/PI.**	10,000	0.329
**SkMel-28 PTX 2_Ed_ PI.**	**10,000**	**0.937**

## Data Availability

Data are contained within the article. The original contributions presented in the study are included in the article; further inquiries can be directed to the corresponding author.

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
