# Peer review of "Morphological and Immunocytochemical Characterization of Paclitaxel-Induced Microcells in Sk-Mel-28 Melanoma Cells"

_biomedicines, 2024, doi:10.3390/biomedicines12071576_

Round 1

Reviewer 1 Report

Comments and Suggestions for Authors

The manuscript presents interesting findings related to microcells, fluorescence photomicrographs are of excellent quality, however, there are several points of concern that need to be addressed.

1, The language and grammar needs lot of improvement, a through editing is suggested.

2. It is clear that authors detected total caspase expression and not the activated caspase status. With such excellent quality of images provided by authors, they can consider staining the cells with fluorescent caspase substrates that will allow them to image status of caspase activation.

3. Figure 6, the cell cycle phase distribution doesn't seem proper, did authors use RNAse prior staining for flow cytometry? Such flat almost normal distribution of cell cycle phase in histogram usually occurs when RNA has not been depleted. Authors should also mention the fixing and permiablization protocol for cell cycle studies. authors acquired too few events for drug treated group, ideally 10000 events should be acquired, if not at least 5000 events, but in the representative figure there are only about 1000 cells generating lot of noise.

4.In discussion section the statement "New studies  indicate that Casp-2, Casp-3, and Casp-6 are not associated with apoptosis but with can-  cerogenesis and activation during differentiation" is simply not correct. Caspase 3 and 6 does play crucial role in intrinsic apoptosis pathway but it is right that under xeno-toxic stress, drug treatment ( specially paclitaxels and platinum drugs) induce non canonical activity of caspases, which involves metastasis, survival etc. Authors should also elaborate the mechanism through which caspases promote carcinogenesis and metastatic potential.

5. line 394 the sentence "Erythroblasts are precursor cells" seems out of context.

Comments on the Quality of English Language

English language editing is needed.

Author Response

Morphological and Immunocytochemical Characterization of Anticancer Drug-Tolerant Cancer Microcell

Morphological and Immunocytochemical Characterization of Paclitaxel-Induced Microcells in Sk-Mel-28 Melanoma Cells

Response to Reviewer 1 Comments

1. Summary

Thank you very much for taking the time to review this manuscript and valuable feedback. I got acquainted with your estimate and questions about the submitted manuscript "Morphological and Immunocytochemical Characterization of Anticancer Drug-Tolerant Cancer Microcell". Please find the detailed responses below and the corresponding revisions/corrections highlighted in the re-submitted files.

2. Questions for General Evaluation

Reviewer’s Evaluation

Response and Revisions

Does the introduction provide sufficient background and include all relevant references?

Yes

Are all the cited references relevant to the research?

Not applicable

Is the research design appropriate?

Yes

Are the methods adequately described?

Can be improved

Are the results clearly presented?

Yes

Are the conclusions supported by the results?

Can be improved

3. Point-by-point response to Comments and Suggestions for Authors

Response 1: Thank you for pointing this out. We agree with this comment. Therefore, we have overread the manuscript and made corrections throughout in text. We used the Writefull app (MS Word) to check corrections.

Comments 2: It is clear that authors detected total caspase expression and not the activated caspase status. With such excellent quality of images provided by authors, they can consider staining the cells with fluorescent caspase substrates that will allow them to image status of caspase activation. 

Response 2: Thank you for pointing this out. Yes, we looked at the total expression of caspases. Thanks for the suggestion, we will take it into account in our future research. However, we cannot currently investigate the features of caspase activation.

Comment 3: Figure 6, the cell cycle phase distribution doesn't seem proper, did authors use RNAse prior staining for flow cytometry? Such flat almost normal distribution of cell cycle phase in histogram usually occurs when RNA has not been depleted. Authors should also mention the fixing and permiablization protocol for cell cycle studies. authors acquired too few events for drug treated group, ideally 10000 events should be acquired, if not at least 5000 events, but in the representative figure there are only about 1000 cells generating lot of noise.

Response 3: In the study, the emphasis is on cells of small size - microcells.  Under the influence of PTX after 48h, the fraction of small cells in the population increases significantly. This is the result of treated cells by PTX. Most small cells are apoptotic cells, but among them, there are also viable microcells, which are recognizable by DNA synthesis using the EdU tagging method. In this study of flow cytometry, one can easily visualize this fraction. However, it is a small fraction but increases significantly after the application of PTX. In addition, the intensity of DNA synthesis is extremely high when related to the amount of nucleic acids in the cell (EdU/PI) (table 2). The intensity of DNA synthesis is more than 20 times higher than the average in the Sk-Mel-28 cell population. This is difficult to explain by the phases of the Sk-Mel-28 cell cycle in the population. 

We have, accordingly, changed Figure 6 to emphasize this point. This figure is reconstructed without using cell size threshold levels. We see that under the influence of PTX, there is a significant increase in small-sized cells among which there are also viable microcells. (Figure 6). Changes can be found in lines 341-346 in the revised manuscript.

Comment 4: In discussion section the statement "New studies indicate that Casp-2, Casp-3, and Casp-6 are not associated with apoptosis but with cancerogenesis and activation during differentiation" is simply not correct. Caspase 3 and 6 does play crucial role in intrinsic apoptosis pathway but it is right that under xeno-toxic stress, drug treatment (specially paclitaxels and platinum drugs) induce non canonical activity of caspases, which involves metastasis, survival etc. Authors should also elaborate the mechanism through which caspases promote carcinogenesis and metastatic potential.

Response 4: Thank you for your opinion and explanation. We have made clarifying changes in this sentence seen in lines 426-427.

We understand your objections, however, Feng et al. [1] illustrated angiogenesis from dying tumor cells in vitro and in vivo mediated by caspase 3. As we understand caspase 3 is a typical apoptotic marker but this study showed caspase is involved in regeneration processes. What are your thoughts on this study?

1. Feng X, Tian L, Zhang Z, Yu Y, Cheng J, Gong Y, et al. Caspase 3 in dying tumor cells mediates post-irradiation angiogenesis. Oncotarget. 2015;6:32353–67.

Comment 5: line 394 the sentence "Erythroblasts are precursor cells" seems out of context.

Response 5: Agree. We have done changes to emphasize this point. We exclude this sentence from our revised manuscript.

6. Response to Comments on the Quality of English Language

Point 1: English language editing is needed.

Response 1: We carefully read the text and made corrections. Our team of authors consists of fluent English speakers.

7. Additional clarifications

Based on the second reviewer's suggestion, we have changed the title of the article “Morphological and Immunocytochemical Characterization of Paclitaxel-Induced Microcells in Sk-Mel-28 Melanoma Cells”. All corrections are marked with a yellow marker.

Sincerely,

Ms. Zane Simsone

Please see the attachment for the response to Reviewer 1

Reviewer 2 Report

Comments and Suggestions for Authors

The manuscript presents a detailed investigation into the morphological and immunocytochemical characteristics of anticancer drug-tolerant microcells, highlighting their potential significance in cancer therapy resistance.

Here are some points that need clarification:

Title:

The title is quite general. Consider revisiting the title to accurately reflect the findings of the study. Here is a suggestion for refining the title: “Morphological and Immunocytochemical Characterization of Paclitaxel-Induced Microcells in Sk-Mel-28 Melanoma Cells.”

 Abstract:

 Rephrase the first sentence to be more accurate: "Biomarkers, including proteins, nucleic acids, antibodies, and peptides, are essential for identifying diseases like cancer and differentiating between healthy and abnormal cells in patients."

Add a sentence at the end to specify the implications for future research or clinical practice: "These findings enhance our understanding of microcell behavior in melanoma, potentially informing future strategies to counteract drug resistance in cancer treatment."

Introduction:

The Introduction provides a lot of background information, but some details might be too specific for this section. Consider focusing on the most relevant points to set up the context of your study. For example:

-The description of drugs (doxorubicin and paclitaxel) could be more concise, focusing only on the aspects directly relevant to the study.

-Discuss all the relevant properties of cancer stem cells in one place rather than spreading them throughout the introduction.

 Materials and Methods:      

 Include a brief description of the Sk-Mel-28 cell line, highlighting its genetic mutations (e.g., BRAF V600E) and relevance to melanoma research. Specifically, the BRAF V600E mutation leads to the activation of the MAPK/ERK signaling pathway, promoting cell growth and survival.

It appears that A549 cells were utilized in at least one experimental context, specifically mentioned in the section discussing nuclear antigen expression after paclitaxel treatment in microcells. However, the exact number of experiments involving A549 cells is not explicitly specified. For clarity and completeness, it would be beneficial for the authors to clearly outline in the Materials and Methods section how many experiments involved the use of A549 cells.

Results:

Discuss the significance of SOX2 and Nanog expression patterns observed in microcells induced by PTX. How do these markers relate to the stem cell-like properties of microcells?

Clarify the significance of PCNA expression in microcells induced by paclitaxel treatment. Discuss how the observed changes in PCNA localization and intensity relate to cell cycle regulation and potential implications for cancer therapy.

Provide a clear comparison between Sk-Mel-28 melanoma cells and A549 lung carcinoma cells regarding PCNA expression patterns in response to paclitaxel. Highlight any differences or similarities observed.

Discussion:

Emphasize the broader implications of microcell presence and behavior in the context of cancer treatment and prognosis. Discuss how these findings contribute to advancing the understanding of tumor resistance mechanisms.

Connect the findings to broader themes in cancer research, such as personalized medicine or the development of targeted therapies.

Consider discussing potential limitations of the study.

Comments on the Quality of English Language

minor editing

Author Response

Morphological and Immunocytochemical Characterization of Anticancer Drug-Tolerant Cancer Microcell

Morphological and Immunocytochemical Characterization of Paclitaxel-Induced Microcells in Sk-Mel-28 Melanoma Cells

Response to Reviewer 2 Comments

1. Summary

Thank you very much for taking the time to review this manuscript and valuable feedback. I got acquainted with your estimate and questions about the submitted manuscript "Morphological and Immunocytochemical Characterization of Anticancer Drug-Tolerant Cancer Microcell". Please find the detailed responses below and the corresponding revisions/corrections highlighted in the re-submitted files.

2. Questions for General Evaluation

Reviewer’s Evaluation

Response and Revisions

Does the introduction provide sufficient background and include all relevant references?

Can be improved

Are all the cited references relevant to the research?

Not applicable

Is the research design appropriate?

Yes

Are the methods adequately described?

Can be improved

Are the results clearly presented?

Can be improved

Are the conclusions supported by the results?

Yes

3. Point-by-point response to Comments and Suggestions for Authors

Comments 1: The title is quite general. Consider revisiting the title to accurately reflect the findings of the study. Here is a suggestion for refining the title: “Morphological and Immunocytochemical Characterization of Paclitaxel-Induced Microcells in Sk-Mel-28 Melanoma Cells.”

Response 1: Thank you for pointing this out. We agree with this comment. Therefore, we deliberated and decided to change the name as per your suggestion. It would be more accurate and describe the essence of this research.

Comments 2: Rephrase the first sentence to be more accurate: "Biomarkers, including proteins, nucleic acids, antibodies, and peptides, are essential for identifying diseases like cancer and differentiating between healthy and abnormal cells in patients."

Response 2: Thank you for pointing this out. We have accordingly changed lines 12-14 to emphasize this point.

Comments 3: Add a sentence at the end to specify the implications for future research or clinical practice: "These findings enhance our understanding of microcell behavior in melanoma, potentially informing future strategies to counteract drug resistance in cancer treatment."

Response 3: Agree. We have added this sentence in lines 25-26.

Comments 4: The Introduction provides a lot of background information, but some details might be too specific for this section. Consider focusing on the most relevant points to set up the context of your study. For example:

-The description of drugs (doxorubicin and paclitaxel) could be more concise, focusing only on the aspects directly relevant to the study.

-Discuss all the relevant properties of cancer stem cells in one place rather than spreading them throughout the introduction.

Response 4: Thank you for your suggestions. We tried to do our best to clarify the doxorubicin and paclitaxel meanings according to the study. The changes can be found in lines 39-46 and 49-51.

We also added information for cancer stem cells in lines 55-56 and 60-70.

Comments 5: Include a brief description of the Sk-Mel-28 cell line, highlighting its genetic mutations (e.g., BRAF V600E) and relevance to melanoma research. Specifically, the BRAF V600E mutation leads to the activation of the MAPK/ERK signaling pathway, promoting cell growth and survival.

Response 5: We added additional information about the Sk-Mel-28 cell line. The changes can be found in the Materials and Methods section in lines 119-120.

Comments 6: It appears that A549 cells were utilized in at least one experimental context, specifically mentioned in the section discussing nuclear antigen expression after paclitaxel treatment in microcells. However, the exact number of experiments involving A549 cells is not explicitly specified. For clarity and completeness, it would be beneficial for the authors to clearly outline in the Materials and Methods section how many experiments involved the use of A549 cells.

Response 6: We understand your opinion and agree suggestions. We added the information about A549 cell line using in our research in the Result section in lines 223-226.

Comments 7: Discuss the significance of SOX2 and Nanog expression patterns observed in microcells induced by PTX. How do these markers relate to the stem cell-like properties of microcells?

Response 7: We added explaining information in the introduction section in lines 99-102 and the discussion section in lines 433-440.

Comments 8: Clarify the significance of PCNA expression in microcells induced by paclitaxel treatment. Discuss how the observed changes in PCNA localization and intensity relate to cell cycle regulation and potential implications for cancer therapy.

Response 8: Thank you for pointing this out. We have accordingly clarified information and added in discussion in lines 455-459 to emphasize this point.

Comments 9: Provide a clear comparison between Sk-Mel-28 melanoma cells and A549 lung carcinoma cells regarding PCNA expression patterns in response to paclitaxel. Highlight any differences or similarities observed.

Response 9: We added observations in the result section in lines 315-320.

Comments 10: Emphasize the broader implications of microcell presence and behavior in the context of cancer treatment and prognosis. Discuss how these findings contribute to advancing the understanding of tumor resistance mechanisms.

Connect the findings to broader themes in cancer research, such as personalized medicine or the development of targeted therapies.

Response 10: Thank you for your opinion and suggestions. We have our vision of microcell significance in personalized medicine in lines 484-492.

Comments 11: Consider discussing potential limitations of the study.

Response 11: Dear reviewer, as well it was reported in our previous research, as in this manuscript in the discussion section in lines 397-400. It has been determined that the number of microcells in cancer tissue is small about 1%. Thus, it can be said that the limiting factor is the number of microcells and their morphological recognition.

12. Response to Comments on the Quality of English Language

Point 1: minor editing

Response 1: Thank you for your suggestion. We have made corrections according to your recommendations. We carefully read the text and made corrections. Our team of authors consists of fluent English speakers.

Sincerely,

Ms. Zane Simsone

Round 2

Reviewer 1 Report

Comments and Suggestions for Authors

Authors have addressed concerns of this reviewer more or less satisfactorily.

Author Response

Morphological and Immunocytochemical Characterization of Paclitaxel-Induced Microcells in Sk-Mel-28 Melanoma Cells

Response to Reviewer 1 Comments

1. Summary

Thank you very much for taking the time to review this manuscript. Please find the detailed responses below and the corresponding corrections highlighted in the re-submitted files.

2. Questions for General Evaluation

Reviewer’s Evaluation

Response and Revisions

Does the introduction provide sufficient background and include all relevant references?

Yes

Are all the cited references relevant to the research?

Not applicable

Is the research design appropriate?

Yes

Are the methods adequately described?

Yes

Are the results clearly presented?

Yes

Are the conclusions supported by the results?

Yes

3. Point-by-point response to Comments and Suggestions for Authors

Comments 1: Authors have addressed concerns of this reviewer more or less satisfactorily.

Response 1: Thank you for pointing this out. We agree with this comment.

4. Response to Comments on the Quality of English Language

Point 1: English language fine. No issues detected

Response 1: We have no objection to the reviewer's rating.

 Sincerely,

Ms. Zane Simsone

Reviewer 2 Report

Comments and Suggestions for Authors

The authors satisfactorily addressed the issues I raised.

I only wanted to point out that lines 484-492 and 494-502 mention the same thing. It is evident that a mistake was made. Lines 484-492 should be removed and an overall conclusion should be written.

Comments on the Quality of English Language

minor editing

Author Response

Morphological and Immunocytochemical Characterization of Paclitaxel-Induced Microcells in Sk-Mel-28 Melanoma Cells

Response to Reviewer 2 Comments

1. Summary

Thank you very much for taking the time to review this manuscript. Please find the detailed responses below and the corresponding corrections highlighted in the re-submitted files.

2. Questions for General Evaluation

Reviewer’s Evaluation

Response and Revisions

Does the introduction provide sufficient background and include all relevant references?

Yes

Are all the cited references relevant to the research?

Not applicable

Is the research design appropriate?

Not applicable

Are the methods adequately described?

Yes

Are the results clearly presented?

Yes

Are the conclusions supported by the results?

Can be improved

3. Point-by-point response to Comments and Suggestions for Authors

Comments 1: The authors satisfactorily addressed the issues I raised.

I only wanted to point out that lines 484-492 and 494-502 mention the same thing. It is evident that a mistake was made. Lines 484-492 should be removed and an overall conclusion should be written.

Response 1: Thank you for pointing this out. We agree with this comment. Therefore, we have made corrections in the revised manuscript in lines 477-486, 489-490 and 493-494.

4. Response to Comments on the Quality of English Language

Point 1: minor editing

Response 1: Thank you for your suggestion. We carefully read the text and made corrections.

Sincerely,

Ms. Zane Simsone